# Depressive Symptoms and Their Associated Factors in Vocational–Technical School Students during the COVID-19 Pandemic

**DOI:** 10.3390/ijerph19063735

**Published:** 2022-03-21

**Authors:** Michele da Silva Valadão Fernandes, Thays Martins Vital da Silva, Priscilla Rayanne e Silva Noll, Alexandre Aparecido de Almeida, Matias Noll

**Affiliations:** 1Federal Institute of Education, Science and Technology of Goiano, Goiás 76300-000, Brazil; thaysvital@iftm.edu.br (T.M.V.d.S.); priscilla.noll@usp.br (P.R.e.S.N.); 2Federal Institute of Education, Science and Technology of Triângulo Mineiro, Minas Gerais 38706-328, Brazil; 3University of São Paulo (Universidade de São Paulo—USP), São Paulo 01246-903, Brazil; 4Federal Institute of Education, Science and Technology of Tocantins, Tocantins 77950-000, Brazil; alexandre.almeida@ifto.edu.br; 5Federal University of Goiás, Goiás 74001-970, Brazil; 6Department of Sports Science and Clinical Biomechanics, University of Southern Denmark, 5230 Odense, Denmark

**Keywords:** COVID-19, adolescents, depression, quality of life, mental health

## Abstract

The objectives of this study were to compare the prevalence of depressive symptoms, behavioral habits, and QoL in students from two vocational–technical schools, and to determine the association of depressive symptoms with behavioral habits and quality of life (QoL) in these students during the COVID-19 pandemic. A cross-sectional survey was conducted with students attending the institution of the Brazilian Federal Network of Professional, Scientific and Technological Education. The students answered a questionnaire on sociodemographic variables, situations related to the COVID-19 pandemic, behavioral habits, QoL, and depressive symptoms. The outcome variable was the presence of depressive symptoms, assessed using the Children’s Depression Inventory (CDI). Poisson regression analysis with robust variance was performed. A total of 343 students participated in this study (women, 55.7%; mean age of 16.1 ± 0.93 years). The prevalence of depressive symptoms among students was 43.4% (95% confidence interval [CI]: 38.0–49.0), and these symptoms were significantly associated with being female (prevalence ratio [PR] 1.72; 95% CI: 1.31–2.27); being in the 10th grade (PR: 1.80; 95% CI: 1.18–2.76) and 9th grade (PR 2.08; 95% CI: 1.37–3.18); social isolation (PR: 2.04; 95% CI: 1.00–4.14); hunger due to a lack of food at home (PR: 1.78; 95% CI: 1.33–2.39); low physical activity levels (PR: 1.68; 95% CI: 1.09–2.59); and moderate (PR: 2.87, 95% CI: 1.68–4.89) and low QoL (PR: 5.66; 95% CI: 3.48–9.19). The results emphasize the importance of interventions aimed mainly at female students and those in the initial years of high school, in addition to the importance of physical activity, food safety, and QoL to improve the mental health of students.

## 1. Introduction

Mental-health-related problems have increased significantly in the last two years. Since social distancing/isolation measures were introduced due to the coronavirus disease (COVID-19) pandemic, caused by the severe acute respiratory syndrome coronavirus 2 (SARS-CoV-2), studies have shown considerable increases in negative symptoms such as depression, anxiety, and stress in people of different ages [1,2]. Adolescents, who are at high risk for mental health problems, began to experience stress, emotional conflicts, fear, and a sudden change in routine more intensely than people in other age groups [3,4]. Among mental health problems, depression was already the most common mental disorder in adolescents, even before the global health crisis [5]. As if the age of vulnerability that is adolescence was not enough by itself, with the pandemic, many of these young people began to experience an increase in anxiety and depressive symptoms, associated with overlapping risk factors [3,4].

Depression is characterized by physical, cognitive, and somatic symptoms, such as a sad or irritable mood, a loss of interest and pleasure, changes in weight and sleep patterns, agitation or motor retardation, fatigue, difficulty concentrating, feelings of worthlessness or guilt, frustration, and suicidal ideation [6,7]. During adolescence, the developmental changes experienced by an individual are marked by emotional instability, which can interfere with positive decision-making [8]. Thus, adolescence is a period of vulnerability for the development of risky mental health behaviors [9,10].

Evidence suggests that depression in adolescents may be associated with risky behaviors such as insufficient levels of physical activity [11,12,13,14], psychoactive substance use [15], unhealthy eating habits [16,17,18], and risky sexual behavior [19,20]. Experts argue that there may be a positive relationship between physical activity and brain health in adolescents, which involves both cognitive function and mental health [14]. Additionally, the use of licit or illicit psychoactive substances is associated with depression in adolescents [9,15]. In adolescence, episodes of rebellion and dangerous habits such as alcohol drinking and drug use may start to develop, especially among adolescents who lack the support of family and friends [9,21]. A lower depressive symptomatology was also found among adolescents who had a good-quality diet, based on the regular consumption of fruits and vegetables, for example [22]. Hence, exploring the associations between depressive symptoms and behavioral habits during the pandemic is important for reducing the high burden related to the increase in mental illness in adolescents.

During the COVID-19 pandemic, prolonged social isolation, grief, domestic violence, and excessive Internet and social media use have contributed to the increase in depressive symptoms [1,23,24,25]. Social isolation and prolonged school closures led to reduced levels of physical activity and increased sedentary habits [26]. Consequently, the COVID-19 pandemic considerably changed the lives of adolescents, increasing the risk of mental illness [27]. In addition, adolescents with more severe depressive symptoms began to experience losses in different dimensions of quality of life (QoL) [2,3,4,28]. Recent studies have already shown that the pandemic has negatively impacted the QoL of adolescents [28]. The psychosocial dimensions of adolescents’ QoL, such as mood, self-esteem, and social relationships, have been affected, evidencing the relevance of the school environment for the promotion of mental health among students [29].

Challenges arising from remote learning during the pandemic affected the teaching–learning process in adolescents [30]. The requirement for mastering new skills by the students, such as proficiency in digital technologies and the ability for autonomous learning, may have increased stress and negatively affected their moods [30,31]. In addition, adolescents, especially vocational–technical school students, are commonly subjected to a heavy study load, academic stress, and excessive activities and school subjects [32,33].

In Brazil, for example, full-time vocational–technical school programs are offered by the Brazilian Federal Network of Professional, Scientific and Technological Education) [34,35]. In the Brazilian educational system, 8.1 million students are enrolled in secondary education, 23.5% of whom are in vocational education; among these, around 500,000 students attend vocational–technical schools [34,35]. Students from vocational–technical schools are subjected to an overload of lessons (around 36–40 h per week), which is much higher than the time that high school students spend studying [36]. However, the available literature on these and other mental health risk factors in vocational–technical school students, especially during the COVID-19 pandemic, remains scarce, thereby limiting our ability to understand and address these problems.

In addition to monitoring depressive symptoms among vocational–technical school students, early diagnosis is crucial for effectively planning public policies that focus on the prevention and treatment of depression in this population [37]. Therefore, it is important to clarify the factors associated with depressive symptoms and to understand the repercussions of the COVID-19 pandemic on the mental health and QoL of these students. Thus, regional differences should be considered as a function of the local culture, study hours, and school environment of the students [38,39,40,41]. Based on the above, the main aim of this study was to compare the prevalence of depressive symptoms, behavioral habits, and QoL in students from two campuses with different socioeconomic and cultural characteristics (urban and rural). Secondarily, our study intended to determine the association between depressive symptoms with behavioral habits (eating habits, psychoactive substance use, risky sexual behavior and physical activity level) and QoL in these students during the COVID-19 pandemic.

## 2. Materials and Methods

### 2.1. Study Type

This study is part of a larger research project, titled “Depressive Symptoms and Quality of Life among Vocational–Technical School Students (SDQV-EMI).” This cross-sectional study was conducted with students of the Brazilian Federal Network of Professional, Scientific and Technological Education from April to July 2021.

### 2.2. Context

Vocational–technical school students from two campuses at an institution of the RFEPCT—IF-Goiano were invited to participate in the study. One of the campuses is located in an urban city (metropolitan area) and the other in a rural city (rural area), in the interior of the state of Goiás [42]. IF Goiano is a multi-campus, multi-curricular, professional, basic, secondary- and higher-education institution, specializing in Vocational Education [43]. The institution offers vocational–technical programs in secondary education in different areas of training, which are structured based on the concept of an integrated curriculum and a comprehensive human training [44].

The vocational–technical school of the Federal Institutes offers full-time programs. However, the workload among the programs may vary with the political pedagogical projects on each campus [45]. Although IF students tend to have a lower average socioeconomic index and greater ethnic–racial diversity, their academic performance is highly similar to that of students from private schools [44]. Their good academic results stem mainly from their general average grades in the National Secondary Education Examination the high-quality training of their teachers, and the good teaching facilities and conditions [44].

The urban city had an estimated population of 22,000 inhabitants, a high Human Development Index (HDI = 0.706), and a per capita Gross Domestic Product (GDP) of approximately BRL 53,000.00 in 2018 [42]. The rural city is located approximately 180 km from the capital Goiânia, has approximately 22,000 inhabitants, and has a high HDI (0.775) [42]. The per capita GDP of the city was BRL 24,893.60 in 2018 [42].

### 2.3. Participants

The sample size was calculated to assess population parameters in the vocational–technical school adolescents from both campuses. Based on the estimated population, the recommended sample size was 257 participants (5% maximum error and 95% confidence level) [46]. The final sample included 343 adolescent students who were 9th, 10th, and 11th graders, that is, who attended the first, second, and third years of the vocational–technical school.

Three hundred and forty-three teenagers aged between 14 and 18 years (55.7% women, with a mean age of 16.1 ± 0.9 years) participated in the study. Of the total sample, 110 participants (73.6% women, 16.1 ± 0.9 years) studied on the campus in the metropolitan area and 233 (47.2% women, 16.0 ± 0.9 years) studied on the campus in the rural area.

### 2.4. Data Collection Procedure

Data were collected using an online questionnaire. The Informed Consent Form (ICF), the Informed Assent Form (IAF), and the research questionnaire were structured and shared via a digital platform. Students who agreed to participate in this study, with the consent of their legal guardians, received the link to the research questionnaire and the digital application via email, as well as their copies of the ICF (guardian) and IAF (minor). The study was approved by the Research Ethics Committee of the Federal Institute of Education, Science and Technology of Goiano (n° 28837120.0.0000.00.36).

### 2.5. Instruments

#### 2.5.1. Sociodemographic- and COVID-19-Pandemic-Related Variables

Sociodemographic (sex, age, color/ethnicity, vocational–technical secondary program, grade, region, access to cell phone/computer, employment status, and household) and COVID-19-pandemic-related (prior diagnosis of COVID-19, social isolation, attitude towards social distancing measures, social distancing measures adopted, and concern with family health and finances and learning during remote classes) data were collected [47].

#### 2.5.2. Behavioral Habits

Behavioral habits encompass issues related to eating habits, psychoactive substance use, risky sexual behavior [21,47,48], and physical activity level. These variables are described below:(A).Eating habits: Feeling hungry due to lack of food in the last seven days; consumption of fruits, vegetables and soda in the last seven days.(B).Psychoactive substance use: Tobacco smoking (smoked at least once; frequency of smoking in the last 30 days); alcohol drinking (drank at least once; frequency of drinking in the last 30 days, experienced getting drunk at least once); illicit drug use, including marijuana, cocaine, and crack (used at least once; frequency of use in the last 30 days).(C).Risky sexual behavior: Prior sexual intercourse; age at first instance of sexual intercourse; number of sexual partners in lifetime.(D).Physical activity level: Self-reported physical activity was assessed using the short version of the International Physical Activity Questionnaire (IPAQ—short version), validated for Brazilian adolescents [48]. The IPAQ—short version assesses the frequency (days/week) and duration (minutes/day) of walking, and moderately and vigorously exercising, in the last seven days. The physical activity level in the participants was classified as “low” (not meeting the criteria for the “moderate” or “high” categories); “moderate” (at least 20 min of vigorous physical activity three or more days/week; at least 30 min of moderate physical activity or walking five or more days/week; any combination of walking, moderate, or vigorous physical activity reaching at least 600 metabolic equivalent of task (MET) minutes/week at least five days/week); and “high” (vigorous physical activity reaching at least 1500 MET minutes/week at least three days/week; any combination of walking, moderate, or vigorous physical activity reaching at least 3000 MET minutes/week at least seven days/week) [49].

#### 2.5.3. Quality of Life

QoL was assessed using the abbreviated version of the World Health Organization Quality of Life Assessment Instrument (WHOQOL-bref), translated and validated into Portuguese [50]. The WHOQOL-bref contains 26 questions scored based on a five-point, Likert-type scale as follows: 1 = “very poor”; 2 = “poor”; 3 = “neither poor nor good”; 4 = “good”; 5 = “very good”. The first two questions refer to self-perception of QoL (WHOQOL-1) and to satisfaction with health (WHOQOL-2). The other 24 items are divided into the following four domains: physical health (7 items), psychological (6 items), social relationships (3 items), and environment (8 items). The four domains of the WHOQOL-bref are converted into a scale ranging from 0–100 points (WHOQOL, 1998). The closer to 100 the total score, the better the QoL. The instrument showed validity, satisfactory internal consistency (Cronbach’s alpha = 0.91), and test–retest reliability [50,51]. For data analysis, QoL indices were categorized into tertiles and classified as “low QoL” (tertile 1—0 to 49.47 points), “moderate QoL” (tertile 2—49.48 to 58.68 points), and “high QoL” (tertile 3—58.69 to 100 points).

#### 2.5.4. Depressive Symptoms

Depressive symptoms were assessed using the version of the Children’s Depression Inventory (CDI) validated for the Brazilian population [52]. The CDI consists of 27 items referring to affective, cognitive, somatic, and behavioral symptoms. Each CDI item consists of three statements scored 0, 1, or 2, and the sum total indicates the final score. The participants had to check the options that best described their situation in the last two weeks. The cut-off score adopted was 17 points [53]. Thus, a score of 17 or higher indicated the presence of depressive symptoms. The version that was validated and standardized by Gouveia et al. [52] had an internal consistency index (Cronbach’s alpha) of 0.81. Other studies conducted with Brazilian adolescents demonstrated good internal consistency, with Cronbach’s alpha ranging from 0.85–0.91 [53,54].

### 2.6. Data Analysis

The data were analyzed using the Statistical Package for the Social Sciences (SPSS), version 25.0, using descriptive statistics and the Chi-squared test (bivariate analysis) to compare the prevalence of depressive symptoms, QoL, and behavioral habits between campuses. The data were tested for normality using the Kolmogorov–Smirnov test. Considering the non-normal distribution of the data, non-parametric tests (Mann–Whitney U Test) were used to compare the median CDI and WHOQOL-bref scores between participants on the two campuses.

Due to the absence of significant differences in the presence of depressive symptoms between the samples from the two campuses, an analysis based on the Poisson regression model with robust variance was performed on the entire sample [55,56]. In this analysis, depressive symptoms were considered outcome variables, while sociodemographic variables, QoL, and behavioral habits were considered independent variables. Poisson regression was adjusted for the following variables, selected according to the literature: sex, age, region, and prior diagnosis of COVID-19 [21,57,58,59,60]. The effect was measured using the prevalence ratio (PR) and corresponding 95% confidence intervals (95% CI) (*p* < 0.05). This method of analysis was chosen because, when the outcome variable has a high frequency, the odds ratio (OR) may overestimate the measure of association [56]. Furthermore, interpreting the OR as risk in cross-sectional studies can be misleading because the prevalence of the study variables is measured, not the incidence [56].

## 3. Results

### 3.1. Sample Description

Table 1 presents the sociodemographic characteristics and COVID-19-pandemic-related situations of the included participants from the two campuses. The populations of students from the urban and rural campuses differed significantly in sex; the urban campus had a greater representation of women, whereas the rural campus was more represented by men (*p* < 0.001).

Among the COVID-19-pandemic-related variables, 30% of all students felt “extremely/very isolated”, 64.1% were “extremely/very concerned” with the health of family/friends, 79% were “extremely/very concerned with their own remote learning”, and 48.4% were “extremely/very concerned with finances”. The measures of social distancing adopted during the pandemic revealed different behaviors between students from the two campuses, with 6.4% of students in the urban city region reporting that they did not follow any social distancing measures, and 40.0% leaving home to perform essential activities, including work. On the rural campus, a higher percentage (50.6%) of students left home to perform essential activities unrelated to work, with 4.7% of students not following any social distancing measures (*p* = 0.037) (Table 1).

### 3.2. Depressive Symptoms, Behavioral Habits and QoL in Students

CDI results indicate a prevalence of 43.4% (95% CI: 38.0–49.0) of depressive symptoms among students. In the sample of the urban campus, the prevalence of depressive symptoms was 43.6% (95% CI: 37.1–49.8). In the rural campus, the prevalence of depressive symptoms was 43.3% (95% CI: 34.6 to 53.0) (Table 2). No differences in eating habits, substance use, or physical activity level were found between campuses. The campuses only differed in prior sexual intercourse and in WHOQOL-bref. Of the students on the urban campus, 32.1% had previous sexual intercourse, whereas on the rural campus, this percentage was 22.7%. A higher percentage of students with high QoL was observed on the rural campus compared with the urban campus (34.3% versus 30.9%, *p* < 0.001).

The average CDI score among students was 15.8 ± 9.4 (95% CI: 14.8–16.8). The analyses showed that female students (17.9; 95% CI: 16.4–19.2) had higher CDI scores than male students (13.1 ± 9.0; 95% CI: 11.7–14.6, *p* < 0.001). The mean CDI scores of the students from the urban (16.40 ± 10.90) and rural (15.47 ± 8.70) campuses did not differ significantly (*p* = 0.922). Female students had higher mean CDI scores than male students, on both the urban (17.73 ± 10.42 versus 12.69 ± 11.57, *p* = 0.009) and rural (17.96 ± 8.43 versus 13.24 ± 8.27, *p* < 0.001) campuses, respectively. In addition, higher mean scores of overall QoL (*p* = 0.004), social relationships (*p* < 0.001), and environment (*p* < 0.001) domains were observed in students from the urban campus (Table 3) than in those from the rural campus.

### 3.3. Depressive Symptoms and Associated Factors during the COVID-19 Pandemic

The results from the Poisson regression analysis are outlined in Appendix A. The results of the adjusted Poisson regression are shown in Figure 1, Figure 2, Figure 3 and Figure 4. Figure 1 shows the adjusted PR results and indicates that the following variables were associated with the presence of depressive symptoms: being female (PR: 1.72; 95% CI: 1.31–2.27); being in the 10th grade/2nd year (PR: 1.80; 95% CI: 1.18–2.76); being in the 9th grade/1st year (PR: 2.08; 95% CI: 1.37–3.18); and experiencing extreme isolation during the pandemic (PR: 2.04; 95% CI: 1.00–4.14).

Of the variables related to behavioral habits, feeling hungry due to lack of food at home rarely (PR: 1.82, 95% CI: 1.43–2.32) and sometimes/often/always (PR: 1.78, 95% CI: 1.33–2.39) in the last seven days were associated with depressive symptoms (Figure 2). Figure 3 indicates that there was no significant association between depressive symptoms and variables related to the use of psychoactive substances and sexual behavior. The prevalence of depressive symptoms was higher among students with low physical activity levels (PR: 1.68, 95% CI: 1.09–2.59) (Figure 4), moderate QoL (PR: 2.87, 95% CI: 1.68–4.89) and low QoL (PR: 5.66, 95% CI: 3.48–9.19) (Figure 4).

## 4. Discussion

This study compared the prevalence of depressive symptoms, behavioral habits, and QoL in students from two campuses with different socioeconomic and cultural characteristics. Although the institutions were located in different geographic regions, no significant differences in the prevalence of depressive symptoms were found between the two groups of students. Except for sexual behavior, behavioral habits were similar between students on both campuses. Differences in sexual behavior between students, specifically regarding greater reports of sexual intercourse among urban campus students, may be related to aspects such as self-efficacy, religiosity, and family support, which are the variables associated with the onset of sexual life among adolescents [61,62]. Sexual initiation, which usually occurs during adolescence, can be an indicator of risk for sexual abuse, early pregnancy, and sexually transmitted diseases [62]. A study of adolescents reported that engaging in sexual activity increased the risk of depression, especially among girls [20]. In addition, other issues involving the number of sexual partners and condom use can directly affect psychological outcomes in adolescents and young adults [20,63,64] Although our findings demonstrated that sexual behavior was not associated with depressive symptoms, future studies should further investigate risky sexual behaviors, especially condom use [20,63], and their impact on the mental health of vocational–technical school students.

Students from the metropolitan area scored higher in overall QoL and in the social relations and environment domains. This difference can be explained by the geographic region of the campus, which reflects the specificities of the local context where students live. Therefore, factors such as social support, interpersonal relationships, and access to recreation, leisure, health services, and transport can determine the self-perceived QoL of these adolescents [28,39]. Other conditions related to individual (gender, age, and grade) and social (interpersonal relationships, sleep habits, and family/school environment) aspects can also affect the level of QoL of students [65]. Social distancing measures were adopted with less caution by students in the metropolitan area; 6.4% of those students did not follow any social distancing measures, whereas 4.7% of students on the rural campus did. This difference may be related to the higher scores in the social relationships and environment domains of students from the metropolitan area, because social isolation restricts activities such as contact with family and friends, opportunities for leisure, and a sense of freedom [23].

In addition, this study evaluated the prevalence of depressive symptoms and their association with behavioral habits and QOL in vocational–technical school students during the COVID-19 pandemic. The prevalence of depressive symptoms in students included in this study was similar to that found among adolescents in some countries in Asia, North America, and Europe, where estimates indicated a high rate of mental illness during the pandemic period [1,66,67]. There was also a significant association between social isolation and depressive symptoms among students. Social deprivation in adolescence may substantially harm emotional health, because adolescents experience a period of increasing need for interaction with peers [68]. Of the adverse conditions experienced by students during the pandemic, the feeling of loneliness has the greatest impact on mental health [31,69]. The feeling of loneliness is a construct strongly associated with depressive symptoms and the social distancing period [70], which may help to explain the higher prevalence of depressive symptoms among students who felt extremely isolated.

Female students had a 1.72 times higher prevalence of depressive symptoms than male students. In addition, women scored higher on the CDI than men on both campuses. This finding is in line with evidence that women suffer more from depression than men [71,72,73]. Several studies have attempted to explain the higher risk of depression among women based on a series of biological and social determinants [72,74]. Among those determinants, five stand out, namely hormonal changes [75], low physical activity levels [76,77], hormonal contraceptive use [78], playing multiple social roles [79,80], and the increased probability of suffering physical and/or sexual violence [81,82]. Therefore, developing preventive actions for the school environment requires a better understanding of these aspects and the development of actions and strategies to positively change the factors that can increase the probability of girls developing depressive symptoms.

The two-fold higher prevalence of depressive symptoms among first year students is a noteworthy finding. Depressive symptoms may decrease over the school years during high school [83] because first year students are adapting to the new educational institution and new interpersonal relationships, which exacerbates negative feelings such as insecurity, fear, anxiety, and stress [84]. Conversely, in the last years, students may be more adapted to the school routine and experience less academic pressure [83]. Other key aspects include profound biological, psychological, and social changes, which are more accentuated in the transition phase between childhood and adolescence [85,86]. These transformations can also considerably affect behavior and emotional control in early high school students [87].

Regarding eating habits, our study showed that students who felt hungry due to a lack of food at home had a higher prevalence of depressive symptoms. Limited or uncertain access to adequate food are conditions related to food insecurity, commonly found in low- and middle-income countries, which have experienced economic recession as a result of the COVID-19 pandemic [88,89]. Food insecurity directly affects the quality of diet and increases the risk of malnutrition and the consumption of unhealthy foods [90,91,92]. Food deprivation, restricted food choice, and anxiety about food supply may directly impair the psychological health of an individual [93]. Therefore, feelings of deprivation and food restriction were likely additional factors that negatively affected the students’ moods during the social isolation period due to the COVID-19 pandemic.

Socioeconomic inequalities, culture, environmental aspects, and access to technologies are known to affect the levels of physical activity in adolescents [77]. In addition, good mental health in adolescents has been associated with high levels of physical activity [12,94,95]. This trend was confirmed in the present study, wherein low physical activity levels were associated with a higher prevalence of depressive symptoms. Mental health in adolescence is directly influenced by their level of physical activity through neurobiological, psychosocial, or behavioral mechanisms [96]. Studies suggest that regular physical activity can contribute to changes in the structural and functional composition of the brain, the opportunity for social interaction, self-regulation, and coping skills among adolescents [97,98]. The regular practice of physical exercise can even increase the synthesis and release of neurotransmitters that improve mood and mental health [99]. Considering that more than half of Brazilian adolescents do not meet the daily recommendation of 60 min of moderate-to-vigorous physical activity, this behavior likely has short- and long-term damaging effects on the physical and psychological well-being of these young people [12,77,100]. Accordingly, effective approaches to combat this issue must target school-age audiences to promote increased physical activity levels and physical and psychological well-being.

Our findings showed that the prevalence of depressive symptoms was associated with moderate and low QoL. It is known that dissatisfaction with QoL is associated with a higher prevalence of symptoms of depression [101,102,103]. Adolescents’ QoL comprises subjective dimensions (psychological, physical, and social functioning) and objective conditions (housing, employment, and educational facilities) [104,105]. However, it can be difficult to specify which aspects of QoL determine psychological health. This is because students’ mental well-being can interact with parameters related to physical activity levels, for example, overweight and obesity [106]. In addition, vigorous physical activities have the potential to indirectly affect QoL by mitigating depressive symptoms [106]. Therefore, further studies are needed to clarify which dimensions of QoL interact with depressive symptoms among vocational–technical school students.

Some limitations may affect the interpretation and generalizability of our results. Because this was a cross-sectional study, we were unable to establish causal inferences. The convenience sample was concentrated in the Brazilian Midwest, precluding the generalization of the findings to the rest of Brazil. Another important limitation results from the online administration of self-reported questionnaires. Online data collection imposed by pandemic restrictions may have limited the sample of participants to those who had internet access. Lastly, we could not exclude residual bias caused by unmeasured variables such as income, anxiety and stress.

This study has several strengths. First, in addition to evaluating the prevalence of depressive symptoms, this study investigated their association with a broad set of indices of behavioral habits and with QoL; these are important variables for identifying students at a higher risk of depression, and have remained overlooked in research during the COVID-19 pandemic. Second, the study was based on a solid methodology using internationally validated instruments and statistical analyses, including the Poisson regression model. Poisson regression is regarded as a robust measure for estimating the prevalence of a given outcome, in terms of proportion [39]. Lastly, we evaluated vocational–technical school students of the Brazilian Federal Network of Professional, Scientific and Technological Education, a research area overlooked in scientific studies. Therefore, our findings help to guide public mental health policies in the context of vocational education, and to contribute to the literature in this field of research.

Based on our findings, future interventions should improve physical activity levels and the conditions of access to regular meals for students. Given the high vulnerability of women, specific approaches to promoting mental health among female students are also needed. Moreover, health and education policies must simultaneously focus on different mental health predictors, especially on increasing social support and providing opportunities for leisure, physical activity and sports to vocational–technical school students.

## 5. Conclusions

Students from both urban and rural campuses showed similar characteristics regarding the prevalence of depressive symptoms and behavioral habits, which differed only in relation to social distancing measures adopted during the COVID-19 pandemic, in sexual behavior, and in self-perception of QoL. This study demonstrated that the high prevalence of depressive symptoms among vocational–technical school students during the COVID-19 pandemic was associated with being female, being in the 10th and 9th grades, experiencing social isolation, and having a moderate or low QoL. Regarding behavioral habits, the prevalence of depressive symptoms was associated with insufficient physical activity and irregular food consumption (feeling hungry) due to the lack of food at home. The results emphasize the importance of interventions aimed mainly at female students and those in the initial years of high school, in addition to the importance of physical activity, food safety, and QoL for the improvement of mental health in students.

## Figures and Tables

**Figure 1 ijerph-19-03735-f001:**
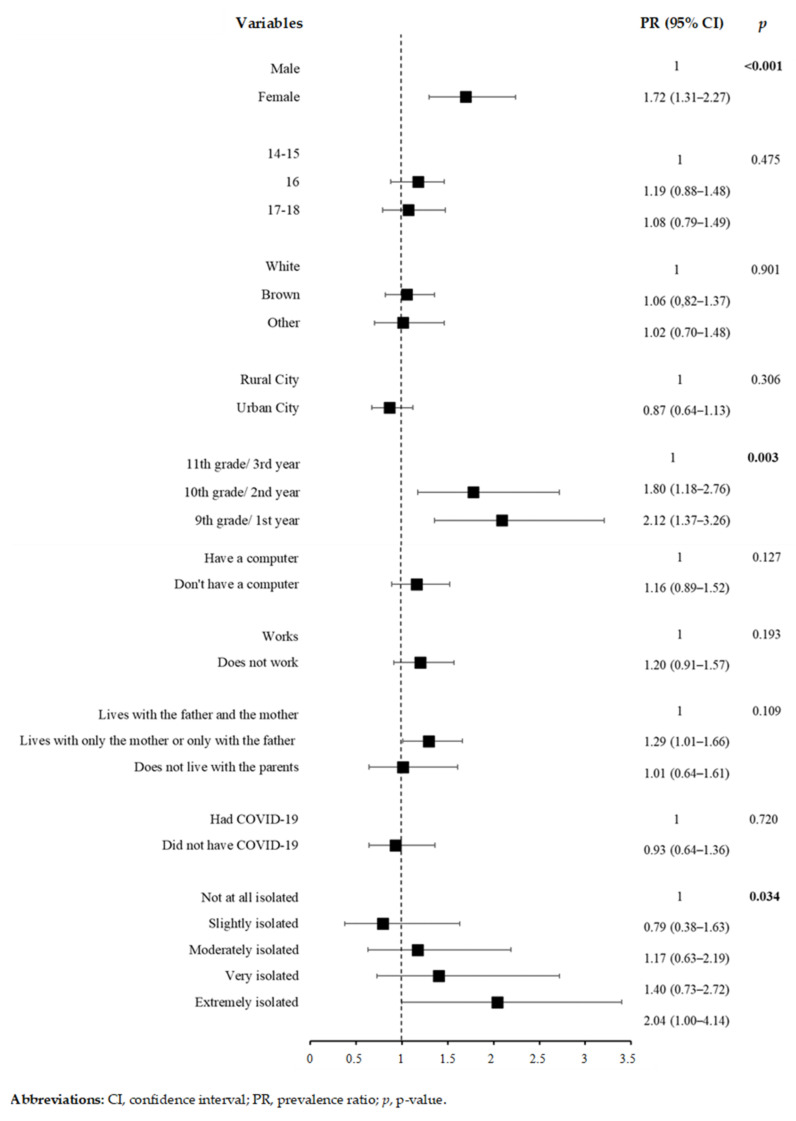
Poisson regression results for depressive symptoms and sociodemographic- and COVID-19-related variables (*n* = 343).

**Figure 2 ijerph-19-03735-f002:**
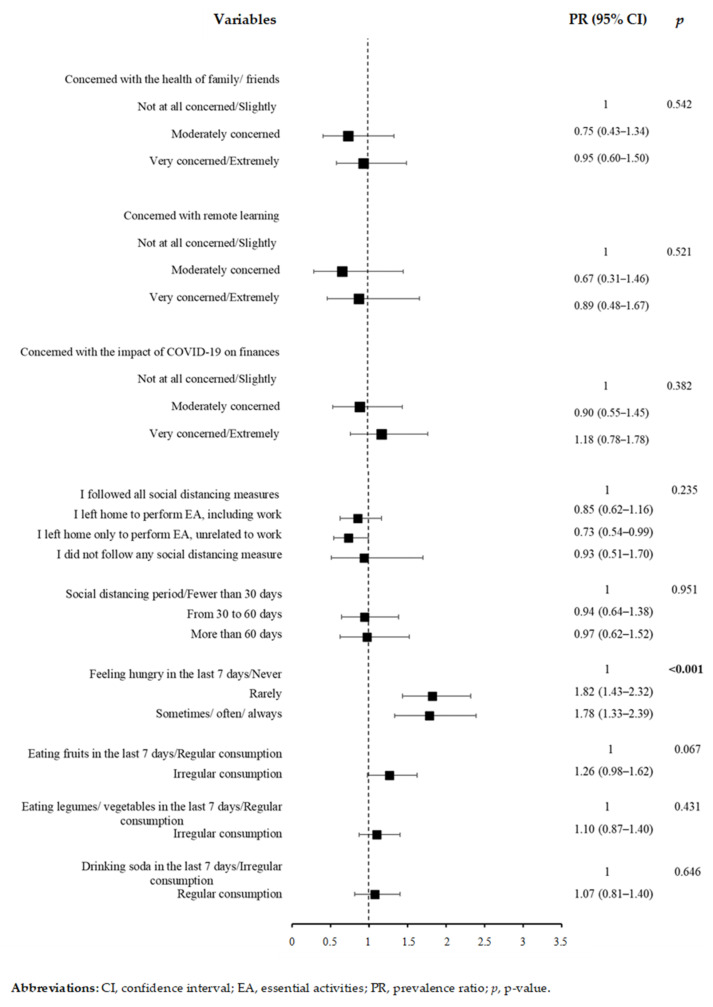
Poisson regression results for depressive symptoms and variables related to COVID-19 and eating habits (*n* = 343).

**Figure 3 ijerph-19-03735-f003:**
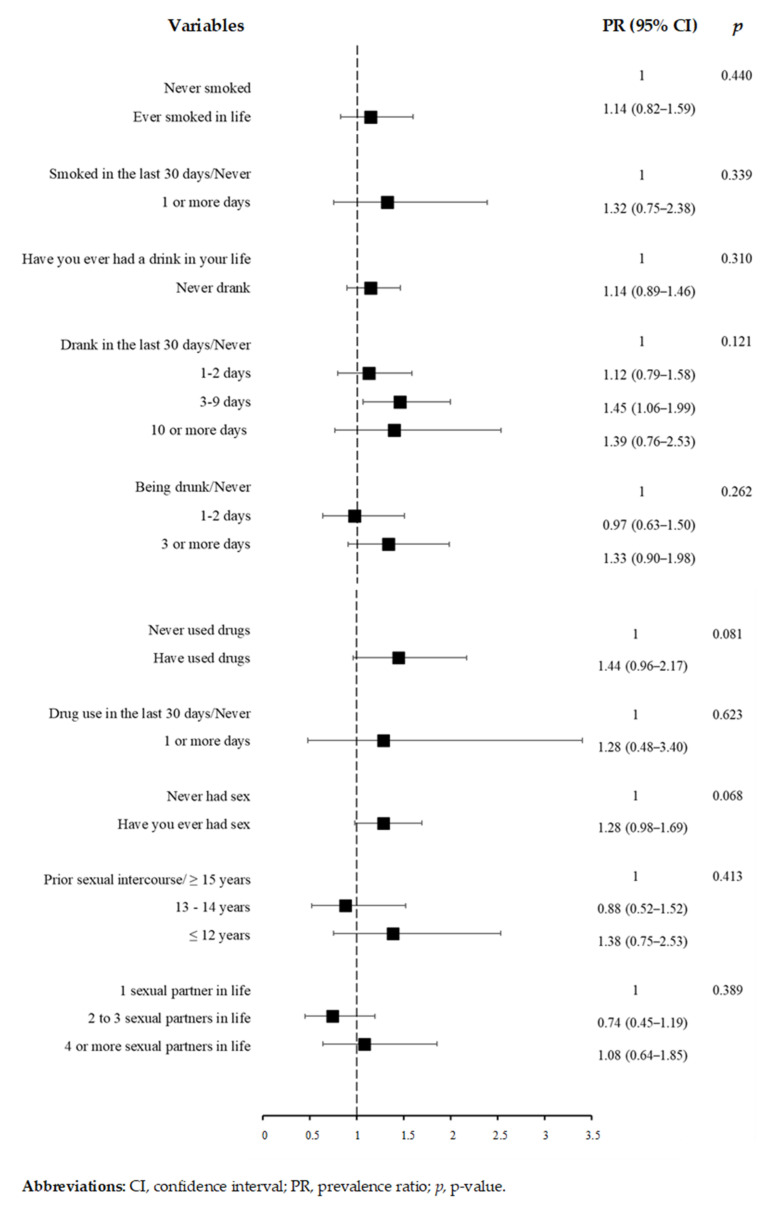
Poisson regression results for depressive symptoms and variables related to psychoactive substance use and sexual behavior (*n* = 343).

**Figure 4 ijerph-19-03735-f004:**
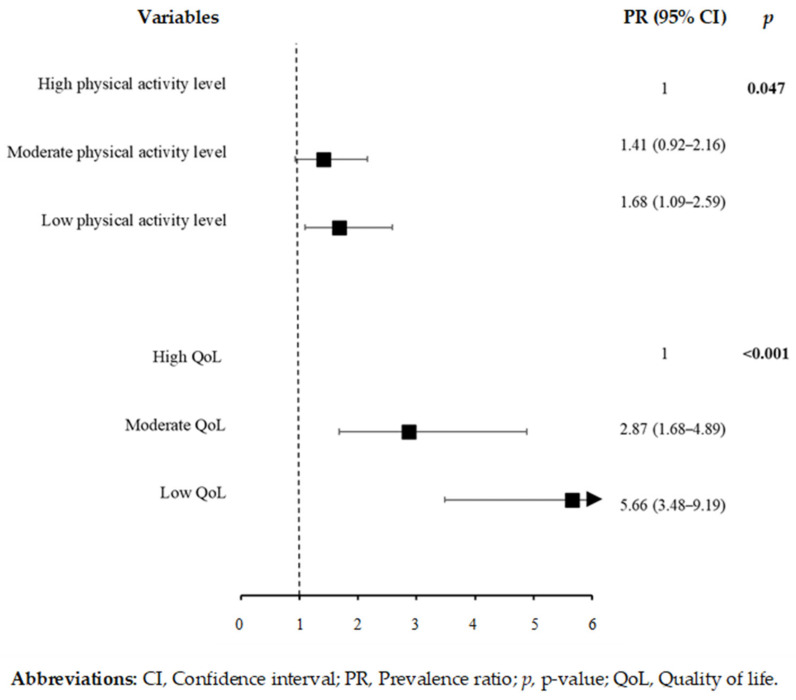
Poisson regression results for depressive symptoms and variables related to physical activity level and QoL (*n* = 343).

**Table 1 ijerph-19-03735-t001:** Sociodemographic characteristics and COVID-19-pandemic-related situations in the sample, stratified by campus.

Variables ^a^		Urban Campus (N = 110)	Rural Campus (N = 233)	
Sociodemographic Characteristics	N (%)	N (%)	N (%)	
Sex				<0.001
Male	152 (44.3)	29 (26.4)	123 (52.8)	
Female	191 (55.7)	81 (73.6)	110 (47.2)	
Age (years)				0.166
14	10 (2.9)	5 (4.5)	5 (2.1)	
15	89 (25.9)	21 (19.1)	68 (29.2)	
16	133 (38.8)	45 (40.9)	88 (37.8)	
17–18	111 (32.4)	39 (35.5)	72 (30.9)	
Color/ethnicity				0.072
White	133 (38.8)	38 (34.5)	95 (40.8)	
Brown	165 (48.1)	51 (46.4)	114 (48.9)	
Other	45 (13.1)	21 (19.1%)	24 (10.3)	
Grade/Year				
9th grade/1st year	150 (43.7)	47 (42.7)	103 (44.2)	0.075
10th grade/2nd year	114 (33.2)	30 (27.3)	84 (36.1)	
11th grade/3rd year	79 (23.0)	33 (30.0)	46 (19.7)	
Computer				0.359
Yes	266 (77.6)	82 (74.5)	184 (79.0)	
No	77 (22.4)	28 (25.5)	49 (21.0)	
Employment				0.520
Yes	83 (24.2)	29 (26.4)	54 (23.2)	
No	260 (75.8)	81 (73.6)	179 (76.8)	
Household				0.121
Lives with the parents	189 (55.1)	57 (51.8)	132 (56.7)	
Lives only with the mother	111 (32.4)	33 (30.0)	78 (33.5)	
Lives only with the father	16 (4.7)	9 (8.2)	7 (3.0)	
Does not live with the parents	27 (7.9)	11 (10.0)	16 (6.9)	
Pandemic-related situations				
COVID-19 diagnosis				0.330
Yes	43 (12.5	11 (10.0)	32 (13.7)	
No	300 (87.5)	99 (90.0)	201 (86.3)	
Social isolation				0.881
Extremely isolated	29 (8.5)	9 (8.2)	20 (8.6)	
Very isolated	69 (20.1)	23 (20.9)	46 (19.7)	
Moderately isolated	142 (41.4)	49 (44.5)	93 (39.9)	
Slightly isolated	68 (19.8)	19 (17.3)	49 (21.0)	
Not at all isolated	35 (10.2)	10 (9.1)	25 (10.7)	
Concern with the health of family/friends				0.269
Extremely concerned	79 (23.0)	32 (29.1)	47 (20.2)	
Very concerned	141 (41.1)	39 (35.5)	102 (43.8)	
Moderately concerned	71 (20.7)	22 (20.0)	49 (21.0)	
Slightly concerned	36 (10.5)	10 (9.1)	26 (11.2)	
Not at all concerned	16 (4.7)	7 (6.4)	9 (3.9)	
Concern with remote learning				0.275
Extremely concerned	165 (48.1)	52 (47.3)	113 (48.5)	
Very concerned	106 (30.9)	29 (26.4)	77 (33.0)	
Moderately concerned	48 (14.0)	17 (15.5)	31 (13.3)	
Slightly concerned	17 (5.0)	9 (8.2)	8 (3.4)	
Not at all concerned	7 (2.0)	3 (2.7)	4 (1.7)	
Concern with the impact of COVID-19 on finances				0.137
Extremely concerned	80 (23.3)	30 (27.3)	50 (21.5)	
Very concerned	86 (25.1)	26 (23.6)	60 (25.8)	
Moderately concerned	91 (26.5)	35 (31.8)	56 (24.0)	
Slightly concerned	51 (14.9)	12 (10.0)	39 (16.7)	
Not at all concerned	35 (10.2)	7 (6.4)	28 (12.0)	
Social distancing measures				0.037
I followed all social distancing measures, and I did not leave home to perform any activity	62 (18.1)	20 (18.2)	42 (18.0)	
I only left home to perform essential activities, unrelated to work	157 (45.8)	39 (35.5)	118 (50.6)	
I left home to perform essential activities, including work	106 (30.9)	44 (40.0)	62 (26.6)	
I did not follow any social distancing measure	18 (5.2)	7 (6.4)	11 (4.7)	
Social distancing period				0.538
Fewer than 30 days	59 (17.2)	19 (17.3)	40 (17.2)	
From 30 to 60 days	94 (27.4)	26 (23.6)	68 (29.2)	
More than 60 days	190 (55.4)	65 (65)	125 (53.6)	

^a^—For some variables, the total scores reflect missing data; *p—p* value of the Chi-square/Fisher’s exact test.

**Table 2 ijerph-19-03735-t002:** Depressive symptoms, behavioral habits, and QoL in the sample, stratified by campus.

Variables ^a^	TOTAL (N = 343) N (%)	Campuses N (%)		*p*
		Urban Campus (N = 110)	Rural Campus (N = 233)	
Eating habits *				
Feeling hungry **				0.696
Never	285 (83.1)	90 (81.8)	195 (83.7)	
Rarely	40 (11.7)	15 (13.6)	25 (10.7)	
Sometimes/often/always	18 (5.3)	5 (4.5)	13 (5.6)	
Eating fruits **				0.714
Regular consumption ^b^	151 (44.0)	50 (45.5)	101 (43.3)	
Irregular consumption	192 (56.0)	60 (54.5)	132 (56.7)	
Eating legumes/vegetables **				0.339
Regular consumption ^b^	212 (61.8)	72 (65.5)	140 (60.1)	
Irregular consumption	131 (38.2)	38 (34.5)	93 (39.9)	
Drinking soda **				0.078
Regular consumption ^b^	100 (29.2)	39 (35.5)	61 (26.2)	
Irregular consumption	243 (70.8)	71 (64.5)	172 (73.8)	
Substance use *				
Tobacco smoking				0.682
Yes	41 (12.0)	12 (10.9)	29 (12.4)	
No	302 (88.0)	98 (89.1)	204 (87.6)	
Smoked ***				0.617
Never	334 (97.4)	106 (96.4)	228 (97.9)	
1–2 days	4 (1.2)	2 (1.8)	2 (0.9)	
3–9 days	1 (0.3)	0 (0.0)	1 (0.4)	
10 or more days	4 (1.2)	2 (1.8)	2 (0.9)	
Alcohol drinking				0.500
Yes	150 (43.7)	51 (46.4)	99 (42.5)	
No	193 (56.3)	59 (53.6)	134 (57.5)	
Drank *** ^c^				0.404
Never	275 (80.2)	85 (77.3)	190 (81.5)	
1–2 days	37 (10.8)	16 (14.5)	21 (9.1)	
3–9 days	23 (6.7)	6 (5.5)	17 (7.4)	
10 or more	8 (2.3)	3 (2.7)	5 (2.2)	
Being drunk *** ^c^				0.617
Never	278 (81.0)	84 (76.4)	194 (83.3)	
1–2 days	37 (10.8)	16 (14.5)	21 (9.0)	
3–9 days	24 (7.0)	9 (8.2)	15 (6.4)	
10 or more days	4 (1.2)	1 (0.9)	3 (1.3)	
Prior drug use ^d^				0.924
Yes	12 (3.5)	4 (3.6)	8 (3.4)	
No	331 (96.5)	106 (96.4)	225 (96.6)	
Drug use *** ^c^				>0.005
Never	340 (99.1)	110 (100.0)	230 (98.7)	
1–2 days	0 (0.0)	0 (0.0)	0 (0.0)	
3–9 days	1 (0.3)	0 (0.0)	1 (0.4)	
10 or more	2 (0.6)	0 (0.0)	3 (0.9)	
Sexual behavior				
Prior sexual intercourse				0.049
Yes	89 (25.9)	36 (32.1)	53 (22.7)	
No	254 (74.1)	74 (67.3)	180 (77.3)	
Age of first sexual intercourse ^c^				
≤12 years	7 (7.9)	3 (8.3)	4 (7.5)	
13–14 years	25 (28.1)	8 (22.2)	17 (32.1)	0.549

≥15 years	57 (64.0)	25 (69.4)	32 (60.4)	
Number of sexual partners in life ^c^				
1	40 (44.9)	18 (50.0)	22 (41.5)	
2 or 3	31 (34.8)	13 (36.1)	18 (34.0)	0.461
4 or more	18 (20.2)	5 (13.9)	13 (24.5)	
Physical activity level				
Low	110 (32.1)	35 (31.8)	75 (32.2)	0.189
Moderate	173 (50.4)	50 (45.5)	123 (52.8)	
High	60 (17.5)	25 (22.7)	35 (15.0)	
WHOQOL-Bref				
High QoL	114 (33.2)	52 (47.3)	62 (26.6)	<0.001
Moderate QoL	115 (33.2)	24 (21.8)	91 (39.1)	
Low QoL	114 (33.5)	34 (30.9)	80 (34.3)	
Depressive symptoms (CDI)				
Yes (≥17)	149 (43.4)	48 (43.6)	101 (43.3)	0.960
No (<17)	194 (56.6)	62 (56.4)	132 (56.7)	

Acronyms: CDI—Children’s Depression Inventory; QoL—Quality of Life; WHOQOL-Bref—World Health Organization Quality of Life. * Classification of eating habits based on indices proposed by the National Survey of School Health (PeNSE); ** Last seven days; *** Last 30 days. ^a^—For some variables, the total scores reflect missing data; ^b^—Regular consumption (at least on 5 of the 7 previous days); ^c^—Only when applicable; ^d^—marijuana, cocaine, crack; *p*—Chi-squared *p*-value/Fisher’s Exact Test.

**Table 3 ijerph-19-03735-t003:** Mean CDI and WHOQOL-bref scores, stratified by campus.

	Total (*n* = 343)	Campus Urban (*n* = 110)	Campus Rural (*n* = 233)	
Variables	Mean (SD), 95% CI	Mean (SD), 95% CI	Mean (SD), 95% CI	*p* *
CDI total score	15.8 (9.4), 14.8–16.8	16.4 (10.9), 14.3–18.4	15.5 (8.7), 14.4–16.6	0.922
Female	17.9 (9.3), 16.4–19.2	17.7 (10.4), 15.4–20.0	18.0 (8.4), 16.4–19.6	
Male	13.1 (9.0), 11.7–14.6	12.7 (11.6), 8.3–17.1	13.2 (8.3), 11.8–14.7	
*p **	<0.001	0.009	<0.001	
WHOQOL-bref				
Physical health	52.6 (14.0), 51.1–54.1	52.8 (16.8), 49.6–55.9	52.5 (18.04), 50.9–54.2	0.913
Psychological	53.0 (19.7), 50.9–55.0	51.1 (22.8), 46.8–55.4	53.9 (18.24), 51.6–56.2	0.243
Social relationships	52.0 (19.5), 49.9–54.0	62.1 (18.2), 58.7–65.6	47.2 (11.8), 44.9–49.6	<0.001
Environment	57.5 (14.6), 56.0–59.1	62.4 (18.3), 58.9–65.8	55.2 (10.8), 53.7–56.8	<0.001
Overall QoL	53.8 (12.8), 52.4–55.1	57.1 (16.0), 54.1–60.1	52.2 (10.8), 50.8–53.6	0.004

Acronyms: CDI—Children Depression Inventory; SD—Standard deviation; CI—confidence interval; QoL—Quality of life; WHOQOL-Bref—World Health Organization Quality of Life. * Mann–Whitney U Test indicating significant differences when *p* < 0.05.

## Data Availability

Due to sensitive data, the data can be accessed upon request to the authors.

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
