# Peer review of "Depressive Symptoms and Their Associated Factors in Vocational–Technical School Students during the COVID-19 Pandemic"

_ijerph, 2022, doi:10.3390/ijerph19063735_

Round 1
Reviewer 1 Report
Dear Authors,
The purpose of the work is good and shows that there is work. This must be appreciated.
However, there are aspects of the work that are especially general and need to be explained in much more detail, so changes are suggested. It is explained in more detail below:
Summary: missing a sentence that concludes the study.
Keywords: between 3 and 5 years.
Introduction:
-Objectives: Properly define the general objective of the study, as well as the specific ones. There is a diversity of aspects that are evaluated and are not specifically included in the objectives.
Method:
- Type of study: Generally, if the study is financed by a project, it is placed at the end of the study and the ethical standards in the procedure section.
- Participants: it is important to detail the sample of participants from the urban city and the rural area. Likewise, the average age, the percentage of each sex, and everything that is known about the sample must be added.
Results:
- Description of participants: the description of participants must be put in the participants section.
- Smaller tables can be made and the different variables can be described (Table 2 and 3).
Discussion:
- The structure of the discussion based on the objectives should be clarified. The structure is confused.
- There is talk of differences between sexual behavior, but it is not said who scores higher and the cause. It only talks about the risks it can pose to adolescents. On the other hand, both in this case and in others, it should be linked to the real objectives of the work.
- Certain details of the quality of life of the residents in the metropolitan area, according to your reflections, could be associated with the self-perceived quality of life. What would be the cause? It must be explained.
- There is talk of the difference between the sexes in terms of depressive symptoms. These analyzes should be improved, and it should also be a key objective in the study. In addition, it would be necessary to define whether what is intended is to make a difference between the sexes or a difference between the urban and rural population, or between both.
- It is claimed that students were predisposed to feel lonely even before the pandemic. Is this something that can be tested?
Citations and references: they must be reviewed based on the regulations determined by the journal.
Reviewer 2 Report
Please see the attachment.

Reviewer 3 Report
Please, rephrase the following for a clearer explanation:
- development of depressive symptoms must be identified
towards understanding repercussions - regional differences should be considered as a
function of the local cultural, study hours and school environment of the students
Fig. 1 is missing.
Round 2
Reviewer 1 Report
Dear authors,
The article has been improved based on the suggestions. This makes the current version more suitable.